# Chemical and Enzymatic Characterization of Leaves from Spanish Table Olive Cultivars

**DOI:** 10.3390/foods11233879

**Published:** 2022-12-01

**Authors:** Eva María Ramírez, Manuel Brenes, Concepción Romero, Eduardo Medina

**Affiliations:** Food Biotechnology Department, Instituto de la Grasa (IG), CSIC, Ctra. Utrera km 1, Building 46, 41013 Seville, Spain

**Keywords:** phenolic compounds, triterpenic acids, polyphenol oxidase, peroxidase, *β*-glucosidase, esterase

## Abstract

Olive leaves are generated as by-products in the olive industry and contain substances with biological properties that provide health benefits. Although these compounds have been characterized in many leaves from olive cultivars devoted to olive oil extraction, few data are available on leaves from the processing of table olives. In this study, the concentration of polyphenols, triterpenic acids, sugars and enzymatic activities (polyphenol oxidase, peroxidase, *β*-glucosidase and esterase) were determined in the leaves of the olive tree (*Olea europaea* L.) of cvs. Aloreña, Cacereña, Empeltre, Hojiblanca, Manzanilla, Verdial, Gordal and Morona. The mean total phenolic content in olive leaves reached 75.58 g/kg fresh weight, and oleuropein was the main polyphenol identified (89.7–96.5%). The main triterpenic acid identified was oleanolic acid, and the main sugar was mannitol, with mean values of 15.83 and 22.31 g/kg, respectively. However, the content of these biocompounds was influenced by the type of cultivar and the orchards of origin. The highest oleuropein content was found in the Manzanilla variety, while the Gordal had the highest triterpene and mannitol content. In particular, the phenolic content could also be affected by endogenous enzymatic activities. High polyphenol oxidase, peroxidase and *β*-glucosidase activity and low esterase activity were detected, compared to the fruit. Similar to the phenolic compounds, enzymatic activities varied with the harvesting season. The lowest phenolic content corresponded to the highest polyphenol oxidase activity detected during spring. The rest of the enzymatic activities also varied throughout the year, but no common trend was observed.

## 1. Introduction

The olive tree (*Olea europaea* L.) has been an autochthonous plant of the Mediterranean basin since ancient times. Nowadays, this crop is spread around the world in regions with a similar climate, and more than ten million hectares of olive trees are cultivated. The olive tree generates a significant quantity of leaves, an agricultural residual biomass considered as a by-product of olive farming and processing. The source of olive leaves comes from the pruning of the trees when unproductive branches are eliminated or when the olive leaves are obtained at olive oil mills and table olive factories. The production of olive residue from pruning has been estimated to be 25 kg per olive tree, and olive leaves from beating olive trees for fruit removal may represent about 5–10% of the total weight of fruit arriving at mills [1,2,3].

These olive residues can be burned in the field or at biorefineries, ground and scattered as vegetal cover on the soil for fertilization purposes, or immediately discarded. Traditionally, these residues have also been employed as animal food, but the residual content of pesticides has limited this use. The olive leaf is also rich in bioactive substances, such as phenolic compounds and triterpenic acids, that could contribute to the valorization of these by-products [4,5,6]. In recent times, a large spectrum of beneficial health properties in vitro and in vivo were attributed to olive leaves and their extracts, related to their bioactive compounds. Phenolic compounds were shown to have significant antioxidant, anti-hypertensive, hypoglycemic, hypocholesterolemic and antimicrobial effects [7]. Promising beneficial properties have been attributed to olive triterpenic acids, such as potent antimicrobial, anti-tumor, anti-inflammatory, and anti-HIV properties, among other activities [8,9]. In fact, extracts of these substances, particularly phenolic compounds, are commercialized by many companies for food additives, dietary supplements, cosmetic and nutraceutical purposes [10].

The main active phenolic constituent in olive leaves is the bitter compound oleuropein, which can constitute up to 6–9% of dry leaf matter [2,4,10]. Other phenolic compounds identified in olive leaves are hydroxytyrosol, the principal degradation product of oleuropein, tyrosol, caffeic acid, *p*-coumaric acid, verbascoside, vanillic acid, vanillin, luteolin, and rutin [4,11,12].

Olive leaves also contain a high concentration of triterpenes, with oleanolic and maslinic acids being predominant among them [4,13]. Triterpenic acids are concentrated on the surface of olive leaves to form a physical barrier that prevents microbes from penetrating the leaf. Fresh olive leaves may contain up to 20 g/kg of triterpenic acids [14]. Furthermore, olive leaves contain a high concentration of sugars, mainly mannitol, along with glucose, sucrose and fructose [2,4]. 

The concentrations of these biocompounds in the leaves depends on several factors, such as origin, olive cultivar, climatic conditions, moisture content, harvesting season, storage conditions, agricultural practices, post-harvest treatment, etc. [1,5,12,15]. Although there is abundant literature on the presence of bioactive compounds in the leaves of olives intended for olive oil production, no data is available on these substances in leaves of olive cultivars destined to be table olives.

As has been commented on above, oleuropein is the major phenolic compound in olive leaves, and it is primarily responsible for the bitterness of the fruits and leaves. The oleuropein chemical structure includes a molecule of elenolic acid linked to hydroxytyrosol by an ester bond, and a molecule of glucose by a glycosidic bond [16]. Oleuropein can be decomposed under the action of light, acids, bases, and high temperature, and by the action of endogenous enzymes in fruit [17] and leaves [18]. 

There are several studies on the presence of these enzymes in fresh fruits, particularly polyphenol oxidase (PPO), which is involved in the browning of harvested olives, and *β*-glucosidase, responsible, to a large extent, for the phenolic profile of olive oil [17,19,20,21,22]. However, few works can be found in the literature aimed at detecting enzymes degrading oleuropein in olive leaves. Motamed et al. [23] followed the changes of peroxidase (POX) and PPO in leaves and buds during olive ripening, and they found an increase in PPO activity with olive maturation. Similar behavior was detected in leaves of the Picual cultivar by Ortega-García et al. [19]. PPO catalyzes the oxidation of phenolic compounds to *o*-quinones, which polymerize into brown pigments that are non-bitter compounds in olives [22,24]. Moreover, *β*-glucosidase can degrade oleuropein during harvesting, storage and damage to the fruits, resulting in oleuropein aglycone and glucose, the former compound being bitter and very soluble in the oily phase of the olives [17,25]. In fruits, researchers detected an increase in *β*-glucosidase activity with maturation [26], and the action of this enzyme during the first months of olive brining is crucial for the debittering of table olives [22]. De Leonardis et al. [21] and Liu et al. [27] reported that *β*-glucosidase activity increased during the storage of olive leaves. Delgado-Povedano et al. [28] investigated the debittering of oleuropein obtained from olive leaves by using commercial *β*-glucosidase, and Paiva-Martins and Pinto [18] reported the formation of oleuropein aglycone in olive leaf extracts. Likewise, hydroxytyrosol can also be formed from oleuropein, or its aglycone, by the action of esterases [27], and this activity increases during the maturation of olives [22,26,29]. However, few studies described the presence and activity of these enzymes in olive leaves, particularly leaves from table olive cultivars.

All these enzymatic activities, that have not previously been studied in olive leaves, could modify the original phenolic profile of the olive leaf, and, consequently, its bioactive capacity. The industrial interest in bioactive substances from natural sources is evident, but the scientific research supporting these claims is lacking. For this reason, the goal of this work was to generate knowledge about the chemical and enzymatic characterization that bioactive substances can undergo in leaves from the main Spanish olive cultivars used for table olive elaboration, and to understand the mechanisms that can affect the profiles of these active compounds.

## 2. Materials and Methods

### 2.1. Raw Material

Samples of olive leaves (*Olea europaea* L.) of Aloreña, Cacereña, Empeltre, Hojiblanca, Manzanilla, Verdial, Gordal and Morona cultivars were hand-harvested randomly from four different orchards located in the provinces of Seville, Córdoba and Málaga (Spain) in October during the 2017, 2019 and 2020 seasons. Leaf samples of Hojiblanca, Manzanilla and Gordal cultivars were also collected in October, December, April and August during the 2020 and 2021 seasons.

All samples were immediately transferred to the laboratory and analyzed on the same day without any storage period. The olive orchards were cultivated under standard cultural practices and trees were irrigated ad libitum to avoid water stress.

### 2.2. Analysis of Phenolic Compounds

The phenolic compounds in the leaves were analyzed as described elsewhere [4]. The leaves were cut into 2–3 mm pieces and 2 g were mixed with 30 mL of dimethyl sulfoxide (DMSO) and homogenized in an Ultra-Turrax homogenizer (Ika, Breisgau, Germany). After 10 min of resting contact, the mixture was centrifuged at 9000× *g* for 5 min. The supernatant (0.25 mL) was diluted with 0.5 mL of DMSO and 0.25 mL of internal standard (0.2 mM syringic acid in DMSO). Samples were filtered through a 0.22 μm pore size nylon filter, and an aliquot (20 μL) was analyzed by high-performance liquid chromatography (HPLC).

The chromatographic system consisted of a Waters 717 plus autosampler, a Waters 600 E pump, a Waters heater module, and a Waters 996 photodiode array detector operated with Empower 2.0 software (Waters Inc., Milford, MA, USA). A Spherisob ODS-2 column (Waters Inc.) at 35 °C and a flow rate of 1 mL/min was used for the analysis. 

The separation was achieved by gradient elution using water (pH 2.5 adjusted with phosphoric acid) and methanol with an initial composition of 90% and 10%, respectively. Gradient elution was described by Ramírez et al. in [17]. Phenolic compounds were monitored at 280 nm. The evaluation of each compound was performed using a regression curve with the corresponding standard. All analyses were performed in duplicate.

### 2.3. Moisture

The water content was determined by weighing 10 g of olive leaves and then oven drying at 105 °C to constant weight. Then, the dried leaves were triturated to a powder using an ultra-centrifugal mill ZM200 (Retsch GmbH, Haan, Germany) for further analysis. All analyses were performed in duplicate.

### 2.4. Analysis of Triterpenic Acids

Extraction of triterpenic acids from leaves was performed as described by Romero et al. [12] with slight modifications. Half a gram of dry leaf powder was mixed with 4 mL of methanol/ethanol (1:1, *v*/*v*), and vortexed for 1 min, centrifuged at 9000× *g* for 5 min at 20 °C, and the solvent was separated from the solid phase. This step was repeated six times, and the pooled solvent extract was vacuum evaporated. Subsequently, the residue was dissolved in 4 mL of methanol and filtered through 0.2 µm pore size. An aliquot (20 µL) was used for HPLC analysis. The chromatographic system and column were the same as those used for the phenolic compound analysis. Elution was performed at 35 °C with a mobile phase of methanol: acidified water with phosphoric acid at pH 3.0 (92:8, *v*/*v*), at a flow rate of 0.8 mL/min and the eluate was monitored at 210 nm. Oleanolic and maslinic acids were quantified using external standards (Sigma, St. Louis, MO, USA). All analyses were performed in duplicate.

### 2.5. Analysis of Reducing Sugars

Sugar compounds and mannitol were analyzed according to the methodology described by Romero et al. [12], with some modifications. One gram of dry leaf powder was mixed with 20 mL of boiling water and 2 mL of internal standard (sorbitol 7.5%). The mixture was vortexed for 1 min, kept for 3 min in an ultrasound bath and vortexed again for 1 min. Then, the paste was centrifuged at 9000× *g* for 5 min and the supernatant was filtered through a paper filter and collected in a 50 mL volumetric flask. This extraction was repeated twice. The solution was kept at 5 °C for 24 h to remove lipids. Two milliliters of the clarified liquid were placed into contact with 1 g of the acidic resin Amberlite IR-120 (FlukaChemieAG, Buchs, Switzerland) and 1 g of the basic resin Amberlite IRA-93 (Fluka). Samples were shaken occasionally for 30 min, and 0.5–1 mL of the solution was centrifuged at 9000× *g* for 5 min and filtered through a 0.22 µm pore size nylon filter. An aliquot (20 µL) was injected into the chromatograph. 

The HPLC system consisted of a Waters 2695 Alliance with a pump, an autosampler and a Waters 410 refractive index detector. A Rezex RCM-Monosaccharide Ca+ (8%) column (300 × 7.8 mm i.d., Phenomenex), held at 85 °C, and deionized water, as eluent, at 0.6 mL/min were used. Standards of sucrose, fructose, glucose, mannitol and sorbitol were employed (Sigma Chemical Co., St. Louis, MO, USA). All analyses were performed in duplicate.

### 2.6. Quantification of Enzymatic Activity

The procedure used to extract the enzymes was based on the methodology described by Ramírez et al. [17]. Acetone powders were prepared by mixing 20–25 g of leaves with 100 mL of cold acetone and polyethylene glycol (2.5 g), previously stored at −20 °C. The residue obtained after vacuum-filtration was re-extracted three times with 100 mL of cold acetone, obtaining a white solid that was dried overnight at room temperature. The acetone powder was stored at −40 °C until use.

Polyphenol oxidase (PPO) and peroxidase (POX) enzymatic extracts were obtained by placing 0.5 g of acetone powder in 20 mL of 0.1 M sodium phosphate buffer with 1 M NaCl, with pH adjusted to 6.2, at 4 °C for 30 min. The suspension was centrifuged at 15,550× *g* for 20 min at 4 °C and the supernatant obtained was the crude enzymatic extract. An aliquot of the supernatant was boiled for 30 min to obtain the denatured enzymatic extract. 

PPO activity was determined by a Shimadzu UV-vis 1800 spectrophotometer at 410 nm (Shimadzu, Kyoto, Japan) [17]. Assays were performed in a mixture of 2.5 mL of 100 mM sodium citrate buffer (pH 5) with 20 mM 4-methylcatechol (Sigma Chemical Co.), as substrate, and 0.5 mL of crude enzyme extract at 25 °C. The assay mixture with the denatured enzyme extract served as the control. One unit of enzyme activity was defined as the amount of enzyme needed to cause an increase in absorbance of 0.05 unit (U) per minute under the conditions mentioned above.

POX activity was determined by spectrophotometry at 470 nm, at 25 °C [17]. The reaction contained 2.7 mL of phosphate buffer, (pH 6) with 40 mM of guaiacol (Sigma Chemical Co.), as substrate, and 26 mM hydrogen peroxide plus 0.3 mL of enzyme extract. The assay mixture with the denatured enzyme extract served as the control. The reaction mixture was incubated for 15 min at 30 °C, and the reaction started with the addition of enzyme extract. One unit of enzyme activity was defined as the amount of enzyme required for an increase of 0.001 unit (U) per minute under the conditions mentioned above. 

The *β*-glucosidase enzymatic extract was obtained by mixing 0.14 g of acetone powder in 10 mL of 10 mM sodium carbonate buffer, containing 5 mM ethylenediaminetetraacetic acid (EDTA), 1 mM phenylmethylsulfonyl fluoride (PMSF), and 1% of 2-mercaptoethanol, all adjusted to pH 9.0 units. The suspension was stirred at 4 °C for 1 h and centrifuged at 15,550× *g* for 20 min at 4 °C. The supernatant was the crude enzyme extract and an aliquot of this was boiled for 30 min to obtain the denatured enzyme extract. The *β*-glucosidase activity was measured by a spectrophotometric method. This activity was determined by monitoring the increase in absorbance at 405 nm related to the increasing amount of *p*-nitrophenol (*p*-NP) liberated from the synthetic glycosidase *p*-nitrophenyl-*β*-D-glucopyranoside (*p*-NPG) (Sigma Chemical Co.). The reaction medium consisted of 100 μL of 50 mM sodium acetate buffer (pH 5.4), 15 mM *p*-NPG, as substrate, and 100 μL of crude enzyme extract. The mixture was incubated at 45 °C and the reaction was stopped by adding 1 mL of 0.5 M sodium carbonate. The assay mixture with the denatured enzyme extract served as the control. The evaluation was performed using a regression curve with *p*-NP in a range of 0–3 mM. One unit of *β*-glucosidase activity was defined as the amount of enzyme required to produce 1 μmol of *p*-NP per minute.

Esterase extract was obtained by mixing 0.25 g of acetone powder suspended in 10 mL of a 10 mM sodium borate buffer (pH 9), containing 5 mM EDTA and 1 mM PMSF. The rest of the extraction procedure was the same as that followed for *β*-glucosidase. 

Esterase activity was carried out according to the method of Ramírez et al. [17]. This activity was determined by continuously monitoring the increase in absorbance at 405 nm at 40 °C related to the increasing amount of *p*-NP liberated from the synthetic *p*-nitrophenyl acetate (*p*-NPA) (Sigma Chemical Co.). The incubation mixture contained 50 μL of crude enzyme extract, 50 μL of 150 mM *p*-NPA in ethanol, as substrate, and 2.9 mL of a 9.2 mM Tris-HCl buffer at pH 7.5. The reaction was initiated by the addition of the enzyme extract. The assay mixture with the denatured enzyme extract served as the control. A standard curve was prepared in the range of 0–0.15 mM of *p*-NP. One unit of enzyme activity (U) was defined as the amount of enzyme required to produce 1 μmol of *p*-NP per minute.

All reactions were carried out in duplicate. The results a#were expressed as a unit of enzyme activity (U) per mg of protein. Protein concentration in the crude enzymatic extract was measured according to the method described by Bradford [30], using the Bradford reagent (Sigma-Aldrich, St. Louis, MO, USA) with crystalline bovine serum albumin (BSA) as the standard protein.

### 2.7. Statistical Analysis

Statistica software 10.0 (StatSoft, Inc., Tulsa, OK, USA) was used for data analysis. Data were expressed as mean values ± standard deviation of duplicates. Statistical comparisons were performed by one-way analysis of variance (ANOVA), followed by Duncan’s multiple range test. A value of *p* < 0.05 was considered statistically significant.

## 3. Results

### 3.1. Chemical Composition of Olive Leaves

The phenolic compounds, triterpenic acids and sugars of the eight cultivars of olive leaves studied are shown in Table 1. 

Oleuropein represented 86.7–96.5% of the total phenolic compounds, being the major compound in olive leaves, with a concentration ranging between 40.06–94.51 g/kg of raw matter (RM). Differences in the oleuropein concentration were observed, both among different varieties and for different orchards within the same variety. The phenolic composition of olive leaves can vary by several abiotic and biotic factors, as described previously [11], among which the type of cultivar has a great influence. The leaves of the Manzanilla variety were those that presented the highest oleuropein concentrations (average value of 79.86 g/kg RM), followed by the Cacereña, Empeltre and Verdial varieties (Appendix A). In contrast, the Gordal variety showed the lowest oleuropein content (50.10 g/kg RM). These differences in the concentration of oleuropein were also found for the fruit, with Manzanilla having the highest concentration, and Gordal showing the lowest concentration [17]. In addition, other phenolic compounds were identified, such as hydroxytyrosol 4-glucoside, hydroxytyrosol 1-glucoside, tyrosol, caffeic acid, verbascoside, ligustroside and luteolin 7-glucoside, in concentrations lower than 3.42–13.25% of the total phenolic content. These compounds were previously studied in olive leaves from trees intended for olive oil production but not in olive leaves from trees for the elaboration of table olives, despite them being harvested before the application of chemical treatments to the tree. These compounds were also characterized in olive leaves of Italian [10,31], Tunisian [32,33], Australian [34] and Spanish [2,12] cultivars, demonstrating, in all of them, high variability of the phenolic content among the different varieties, even within the same cultivar. The eight table olive cultivars analyzed in this study showed a concentration of phenolic compounds, in particular oleuropein, higher than those found for varieties previously studied by other authors. Majetić Germek et al. [35], Medina et al. [4] and Lama-Muñoz et al. [2], found ranges in oleuropein concentration between 38.18–105.72, 44.79–108.28 and 43.40–122.30 g/kg, respectively, on a dry weight basis. If we take into account an average humidity of around 45% in the leaf, these concentrations are almost half those found in our study. This fact could be explained by the fact that, in the previous studies, the samples were analyzed from dry leaves (dried at room temperature or heating). Nevertheless, our analysis was carried out on fresh leaves, directly collected, without any drying treatment or storage. During the drying process, cell tissues break down and the phenolic compounds come into contact with endogenous enzymes (PPO, POX, *β*-glucosidase and esterase) that can lead to the hydrolysis and oxidation of these compounds [17,18,19,20,21,22,23,24,25,26,27,28,29]. This would explain the presence of hydroxytyrosol, a product of the hydrolysis of oleuropein, in olive leaves from dried samples in some studies. Likewise, the different extraction procedures used could obtain different yields, well-described by Lama-Muñoz et al. [2]. Our results were more similar to those obtained by Romero et al. [10], who found oleuropein concentrations between 47.46–72.11 g/kg RM for the fresh leaves of the Picual and Arbequina cultivars, whose fruits are devoted to olive oil extraction, and that were analyzed directly after collection.

Olive leaves are also rich in triterpenic acids (Table 1), including a high concentration of oleanolic acid, that was found in all the analyzed samples. The Hojiblanca, Gordal and Manzanilla leaves were the varieties with the highest oleanolic acid content, with average concentrations of 17.47, 17.26 and 16.77 g/kg RM, respectively (Appendix A). Maslinic acid demonstrated lower average concentrations, between 2.51 g/kg RM for the Aloreña variety and 3.73 g/kg RM for the Empeltre variety. Differences in triterpene content were also observed among different varieties and samples within the same variety, although these variations were much lower than those found for phenolic compounds. This variability for different cultivars, and the prevalence of oleanolic acid versus maslinic acid, was also previously observed by other researchers in studies carried out with olive leaves of Spanish varieties of Arbequina, Hojiblanca, Lechín and Picual cultivars [13,14], and among commercial leaves [4]. Surprisingly, this ratio of triterpenes is reversed in fresh olives, in which maslinic acid is the main triterpene. This phenomenon was studied by Romero et al., finding that the concentrations of maslinic and oleanolic acids did not accumulate in the same proportions in the leaves as in the fruits of the same tree [12].

In addition, the sugar content in olive leaves was characterized (Table 1), with the mannitol being the main compound, presenting a minimum concentration of 16.29 g/kg RM for the sample Manzanilla (orchard 1) and a maximum concentration of 25.98 g/kg RM for the sample Gordal (orchard 2). These results were consistent with those found for the Picual, Arbequina, Royal and 3 wild varieties [3,12], and samples of commercial olive leaves [4]. Mannitol is a polyalcohol naturally found in many plants. Based on its beneficial physiological effects, mannitol is currently used as a sugar-free functional sweetener in the food industry and has great potential in the medical and food industries, increasing its demand [36]. The presence of mannitol in olive leaves makes them a promising source of this bioactive compound. In addition, other sugars were identified, such as sucrose, glucose and fructose, with mean concentrations for all the varieties analyzed of 5.01, 5.85 and 3.81 g/kg, respectively (Appendix A). These values were in line with previous studies [4,12], in which the cultivars and the harvesting seasons influenced the quantities of sugars.

The content of bioactive compounds in the leaves can be influenced by different factors, such as seasonal changes and the ripening of the fruits [15,37]. Figure 1 shows the changes in the content of phenols, triterpenes and sugars over time, in the leaves of three varieties of olive trees. April was the month in which oleuropein concentrations were lower with values of 49.88, 31.08 and 50.61 g/kg RW for Manzanilla, Gordal and Hojiblanca, respectively, which were statistically different from the values found for the rest of the time-frames (Figure 1A). However, in August, the concentration of oleuropein increased and remained constant until October, before harvesting the fruit. Subsequently, the oleuropein content decreased slightly in Manzanilla and Gordal leaves (no significant differences) and increased in Hojiblanca in December, after harvesting the fruit. The changes in the phenolic content were independent of the moisture in the leaves, since that remained constant throughout the different seasons. This trend was previously observed by Ortega-García et al. [19] for the Picual, Verdial, Arbequina and Frantoio varieties, with the maximum concentrations of total phenols being prevalent in the month of July. Mitsopoulos et al. [37] also observed, for ten Greek varieties, that the maximum concentrations of phenolic compounds were reached in September, with the month of April being when the concentrations were lowest. This variation was also observed for Picual leaves, reporting a higher concentration of oleuropein during the cold season [11,12].

The concentration of triterpenic acids was also influenced by the harvesting time, although the differences in concentration were less than those in phenolic compounds. December and April were the months with significantly lower total concentrations for the three varieties studied (Figure 1B), as was the case with oleuropein. Exactly the opposite occurred in August and October, when the maximum concentrations for these compounds were observed. Romero et al. [12] also observed this difference in the leaves of the Picual variety collected in April and November. Regarding the sugar content, no significant differences were observed for the Manzanilla and Gordal cultivars over time. Only an increase, occurring in October, for the Hojiblanca variety was observed (Figure 1C).

### 3.2. Enzymatic Activity in Olive Leaves

The endogenous enzymes present in olive leaves can act on polyphenols, specifically on oleuropein, the major compound, modifying the phenolic profile and concentration [21], but there is very little information about the enzymatic activity in olive leaves, particularly in those leaves of cultivars intended for table olive elaboration [19,23].

Figure 2 shows the activity of the oxidase enzymes, polyphenol oxidase (PPO) (Figure 2A) and peroxidase (POX) (Figure 2B) of the eight varieties of leaves studied. The highest PPO activity was detected in the leaves of the Empeltre variety, followed by Manzanilla and Gordal, with mean values of 109.13, 88.46 and 85.05 U/mg of protein, respectively. However, great variability was observed within the samples belonging to the same variety with differences of more than double in activity, as seen for the two samples studied of the Empeltre variety. In olives, the PPO enzyme is involved in the oxidation of phenolic compounds, giving rise to brownish-colored compounds as a product, responsible for the darkening of the fruit [24], as well as the appearance of dark spots, caused by damage occurring during harvesting [17]. In this latter study, the fruit of the Manzanilla variety showed a higher PPO activity, and, consequently, a greater appearance of dark spots.

Olive leaf peroxidase activity reached higher values compared to PPO (Figure 2B). The Aloreña, Cacereña and Gordal varieties had the highest POX activity (76,819.79, 69,757.13 and 69,631.16 U/mg of protein, respectively). The rest of the cultivars showed lower activities, with the Empeltre variety being the one with the lowest average activity, in contrast to the PPO. The POX activity was much higher in olive leaves than that recorded in previous studies carried out with table olives [17]. This high POX activity could influence the oxidation processes of phenols in leaves, although its contribution to oxidation could be limited by the availability of H_2_O_2_ [20].

The high variability found for the PPO and POX activities was evident among different varieties and different samples within the same variety, as occurred in fresh olive fruits of the Gordal, Manzanilla and Hojiblanca varieties [17]. The leaf harvesting season also influenced the oxidoreductase activity of the leaves (Appendix A). In this way, the maximum PPO activity was reached in April for the three varieties studied (Appendix A), together with the lowest phenolic content of the samples (Figure 1). In contrast, the PPO activity was significantly lower in December for the Manzanilla and Gordal varieties, while the lowest activity for the Hojiblanca variety was in August and October. These results were in line with those obtained by Ortega-García et al. [19] in leaves of the Picual variety, in which the PPO activity increased during fruit ripening, as did the concentration of oleuropein. The POX activity for the Gordal and Hojiblanca varieties was higher in August when the concentration of oleuropein in the leaves was higher (Appendix A). Motamed et al. [23], showed that both PPO and POX activity in leaves of the Zard variety increased during ripening and decreased with fruit softening. In contrast, the Manzanilla variety demonstrated a different behavior, with April being when the highest POX activity was recorded, at the same time as the highest PPO activity and lowest phenolic content.

The presence of hydrolytic enzymes was also determined in the leaves, specifically the *β*-glucosidase (Figure 3A) and esterase activities (Figure 3B). Again, the high variability between the different varieties, and between samples within the same variety, was evident. The leaves of the Empeltre and Gordal varieties showed the highest mean values of *β*-glucosidase activity (231.33 and 167.58 U/mg of protein, respectively), while the Cacereña and Morona varieties had the lowest activities (46.40 and 53.30 U/mg of protein). The esterase activity is shown in Figure 3B. The Morona variety was the one with the highest activity (0.80 U/mg of protein), followed by the Aloreña, Manzanilla and Verdial varieties, although with activity values below 1.0 U/mg of protein in all cases; so, the absence of esterase activity can be considered in these samples.

Previous studies reported that the hydrolysis of oleuropein from the leaves was associated with the action of *β*-glucosidase [18,28]. High *β*-glucosidase activity is related to a greater transformation of oleuropein into its aglycone, which is still a bitter compound. The presence of esterase enzymes could hydrolyze the ester bond of this molecule and release the hydroxytyrosol molecule (not bitter), a compound with a high antioxidant capacity [16]. The concentration of hydroxytyrosol in olive fruit, due to esterase activity, increases with the ripening of the fruit [17]. The low values of esterase activity, compared to the fruit, supports the non-detection of hydroxytyrosol in the leaves of the different varieties analyzed.

## 4. Conclusions

The olive industry generates a high quantity of leaves that contain valuable bioactive substances, the recovery of which could contribute to the valorization of these waste products. This study determined the composition of phenolic compounds, triterpenic acids and sugars in olive leaves from the main Spanish varieties intended as table olives. This study detected great variability in the content of these compounds, depending on the cultivar analyzed, even though the differences were evident among samples of the same variety. The olive leaf composition showed a high concentration of oleuropein, oleanolic acid and mannitol, which are considered to be important sources of bioactive compounds. While the concentrations of triterpenic acids and sugars remained very similar throughout the year, the phenolic compounds were affected by the harvesting season, with the lowest concentrations recorded in April. Similarly, these phenolic compounds could be affected by the various enzymatic activities in the leaf. While the concentration of oleuropein was the lowest in April, the PPO activity was higher for all varieties. In addition, a high POX and *β*-glucosidase activities were detected, which could influence the degradation of oleuropein and its availability in the leaves. These enzymatic activities also varied throughout the year, although a common trend was not observed.

These results contribute to the knowledge of the presence of bioactive substances in olive leaves of table olive varieties, and help in the selection of olive leaf varieties, based on the content of beneficial compounds for health.

## Figures and Tables

**Figure 1 foods-11-03879-f001:**
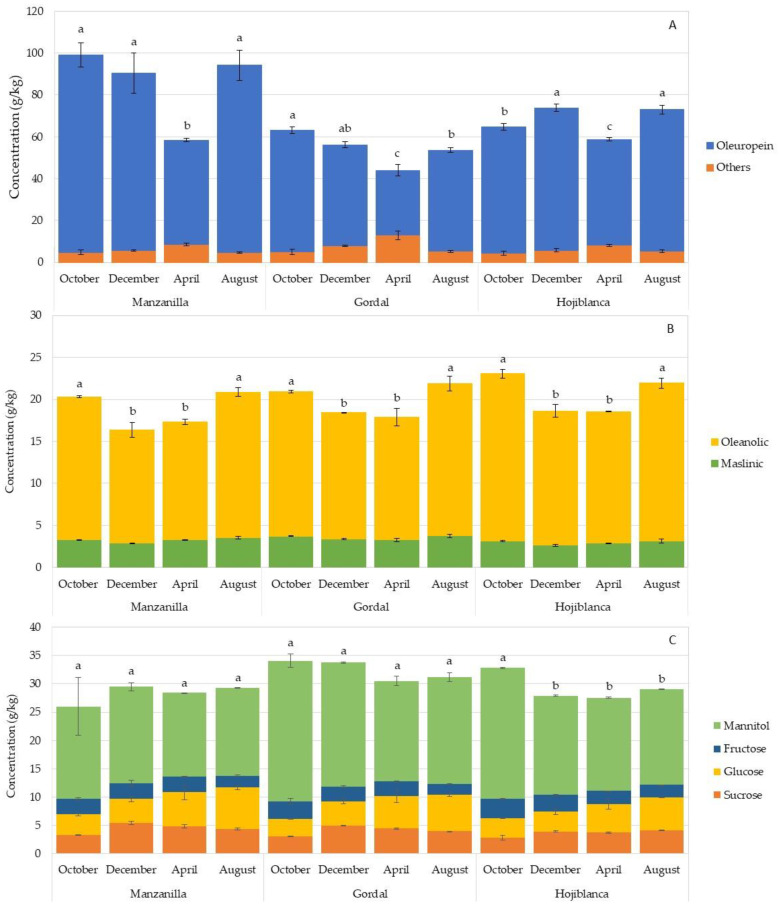
Phenolic compounds (**A**), triterpenic acids (**B**) and sugars (**C**) (g/kg) in olive leaves of the Manzanilla, Gordal and Hojiblanca varieties collected from October to August during the 2020/2021 season. “Others” is the sum of hydroxytyrosol 4-glucoside, hydroxytyrosol 1-glucoside, tyrosol, caffeic acid, verbascoside, ligustroside and luteolin 7-glucoside. Data are means of duplicates. Vertical bars mean the standard deviation. Different lowercase letters indicate significant differences among harvesting times for each variety according to Duncan’s multiple range test (*p* < 0.05).

**Figure 2 foods-11-03879-f002:**
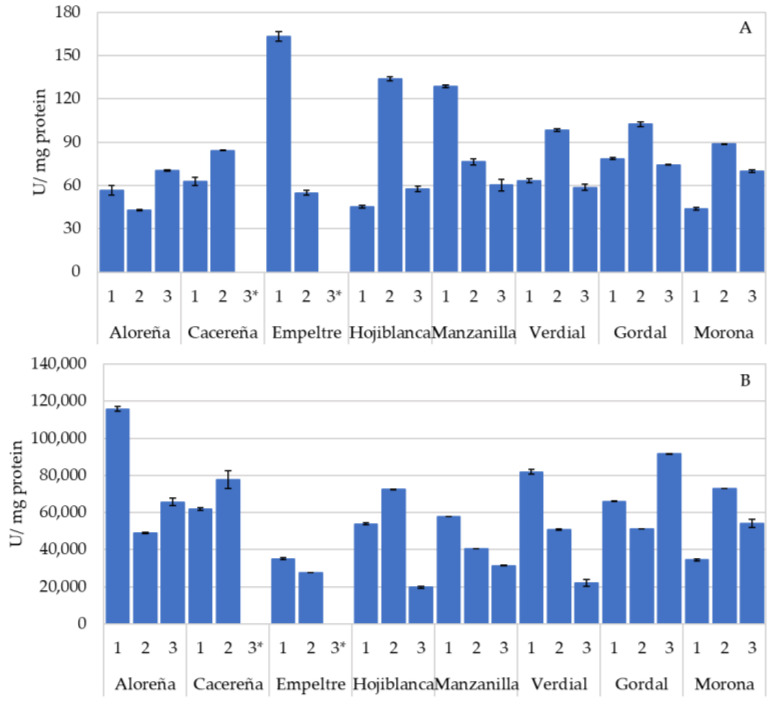
Oxidase enzymatic activity (U/mg protein) in olive leaves of Aloreña, Cacereña, Empeltre, Hojiblanca, Manzanilla, Verdial, Gordal and Morona varieties. Panel (**A**) corresponds to polyphenol oxidase activity. Panel (**B**) corresponds to peroxidase activity. Data are means of duplicates. Vertical bars mean the standard deviation. * Data not available.

**Figure 3 foods-11-03879-f003:**
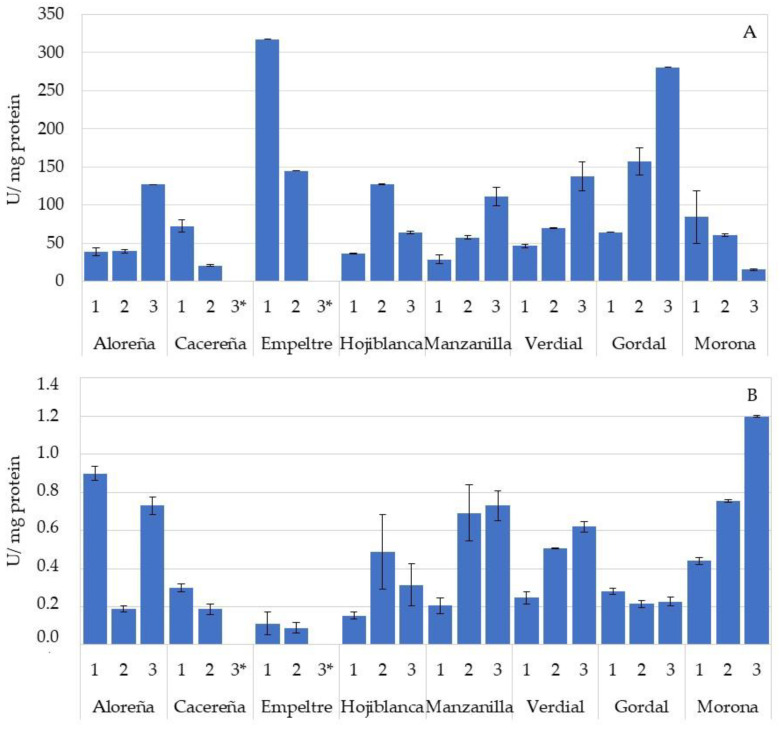
Hydrolase enzymatic activity (U/mg protein) in olive leaves of Aloreña, Cacereña, Empeltre, Hojiblanca, Manzanilla, Verdial, Gordal and Morona varieties. Panel (**A**) corresponds to *β*-glucosidase activity. Panel (**B**) corresponds to esterase activity. Data are means of duplicates. Vertical bars mean the standard deviation. * Data not available.

**Table 1 foods-11-03879-t001:** Concentration of phenolic compounds, triterpenic acids and sugars (g/kg) in the olive leaf samples from 4 different orchards. Data are expressed as the mean values of duplicates. Standard deviation is in parenthesis. “Others” is the sum of hydroxytyrosol 4-glucoside, hydroxytyrosol 1-glucoside, tyrosol, caffeic acid, verbascoside, ligustroside and luteolin 7-glucoside.

Variety	Orchard	Phenolic Compounds		Triterpenic Acids		Sugars			
		Oleuropein	Others	Maslinic Acid	Oleanolic Acid	Sucrose	Glucose	Fructose	Mannitol
**Aloreña**	1	55.06 (2.67)	6.23 (0.56)	2.80 (0.02)	15.11 (0.08)	3.57 (0.36)	4.22 (0.52)	3.15 (0.32)	22.76 (0.10)
	2	62.77 (2.06)	9.59 (0.84)	3.06 (0.22)	14.43 (2.08)	7.18 (0.27)	4.57 (0.34)	4.01 (0.13)	22.12 (0.06)
	3	67.59 (4.51)	4.98 (0.78)	2.32 (0.10)	15.38 (1.08)	14.37 (0.05)	5.40 (0.61)	3.10 (0.51)	23.51 (0.98)
	4	64.29 (2.96)	3.34 (0.31)	1.84 (0.27)	13.92 (2.08)	7.38 (1.43)	8.57 (1.27)	4.51 (0.08)	22.78 (1.91)
**Cacereña**	1	66.13 (3.78)	4.51 (0.18)	4.02 (0.09)	12.36 (0.08)	3.97 (0.20)	3.42 (0.17)	3.37 (0.33)	22.76 (1.04)
	2	83.08 (3.98)	6.60 (1.12)	2.33 (0.20)	18.26 (2.08)	4.88 (0.77)	5.86 (0.65)	3.33 (0.17)	21.97 (0.46)
	3	79.53 (7.76)	2.97 (0.62)	2.27 (0.41)	10.53 (4.08)	6.98 (0.59)	9.49 (2.42)	4.28 (0.57)	21.58 (0.77)
**Empeltre**	1	75.12 (1.12)	6.01 (0.75)	4.30 (0.38)	16.97 (3.08)	4.30 (0.47)	3.83 (0.47)	3.13 (0.23)	25.24 (2.37)
	2	65.85 (8.55)	6.33 (0.21)	3.91 (0.06)	16.79 (0.08)	3.93 (0.44)	3.47 (0.45)	3.95 (0.36)	21.66 (0.30)
	3	76.54 (0.52)	4.34 (0.56)	2.97 (0.14)	13.41 (1.08)	4.74 (0.36)	8.46 (0.44)	4.27 (0.17)	24.92 (0.09)
**Hojiblanca**	1	60.41 (1.58)	4.87 (0.43)	3.14 (0.10)	19.91 (1.08)	2.79 (0.39)	3.49 (0.01)	3.44 (0.00)	23.06 (0.15)
	2	77.00 (17.96)	6.74 (0.08)	2.71 (0.41)	18.83 (4.08)	2.07 (0.01)	3.36 (0.72)	3.12 (0.19)	18.39 (1.68)
	3	57.82 (1.20)	3.47 (0.12)	2.97 (0.03)	16.71 (0.08)	4.18 (0.13)	3.15 (0.22)	3.46 (0.39)	21.22 (0.06)
	4	67.57 (6.03)	3.52 (0.20)	1.77 (0.32)	14.45 (3.08)	4.76 (2.36)	7.93 (0.31)	4.13 (0.07)	19.68 (0.34)
**Manzanilla**	1	94.51 (5.74)	4.70 (1.12)	3.33 (0.02)	16.98 (0.08)	3.30 (0.10)	3.68 (0.32)	2.70 (0.24)	16.29 (5.11)
	2	71.60 (10.38)	4.08 (0.20)	3.36 (0.05)	18.13 (0.08)	4.44 (0.53)	5.76 (0.98)	3.75 (0.75)	23.75 (0.40)
	3	85.92 (10.18)	7.70 (0.54)	3.33 (0.85)	16.65 (8.08)	4.47 (0.24)	5.17 (0.25)	3.85 (0.18)	19.61 (0.22)
	4	67.41 (0.79)	2.56 (0.23)	2.57 (0.24)	15.30 (2.08)	5.02 (0.15)	10.15 (1.13)	5.03 (0.21)	19.42 (2.14)
**Verdial**	1	74.86 (3.42)	4.35 (0.40)	3.15 (0.06)	13.29 (0.08)	4.25 (0.45)	3.63 (0.65)	3.68 (0.50)	19.53 (1.51)
	2	80.61 (0.30)	5.19 (0.18)	3.40 (0.27)	18.68 (2.08)	2.95 (0.24)	3.53 (0.21)	3.79 (0.07)	24.04 (1.25)
	3	61.83 (4.69)	5.61 (0.69)	3.41 (0.02)	16.04 (0.08)	8.01 (0.17)	6.34 (0.25)	3.20 (0.06)	21.30 (0.91)
	4	67.60 (3.38)	4.24 (0.26)	1.96 (0.08)	11.00 (0.08)	5.09 (0.33)	10.99 (0.31)	4.93 (0.88)	19.32 (0.85)
**Gordal**	1	58.17 (1.60)	4.90 (1.30)	3.72 (0.04)	17.21 (0.08)	3.05 (0.01)	3.08 (0.10)	3.11 (0.51)	24.79 (1.19)
	2	45.62 (5.78)	5.73 (0.17)	3.89 (0.77)	18.90 (7.08)	4.67 (0.22)	4.72 (0.17)	3.04 (0.38)	25.98 (0.75)
	3	40.06 (6.87)	5.98 (0.59)	4.88 (0.19)	19.15 (1.08)	4.01 (0.28)	4.09 (0.09)	3.97 (0.02)	24.32 (0.13)
	4	56.55 (17.86)	3.69 (1.62)	2.21 (0.25)	13.76 (2.08)	4.40 (0.34)	6.50 (0.47)	3.89 (0.33)	23.59 (1.38)
**Morona**	1	69.24 (3.58)	4.90 (1.21)	2.91 (0.13)	16.76 (1.08)	4.78 (0.11)	7.15 (0.38)	4.49 (0.41)	23.53 (0.91)
	2	74.20 (10.38)	3.92 (0.80)	2.42 (0.05)	14.37 (0.08)	5.24 (1.45)	6.70 (0.23)	3.89 (0.05)	20.89 (0.28)
	3	67.10 (8.23)	3.39 (1.67)	3.79 (0.03)	19.08 (0.08)	6.45 (0.09)	5.70 (0.38)	4.43 (0.39)	25.83 (0.30)
	4	45.34 (3.12)	1.60 (0.55)	2.32 (0.55)	14.91 (5.08)	5.35 (0.93)	13.25 (0.49)	5.58 (0.64)	24.18 (2.37)

## Data Availability

The data is contained within the article.

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
