# Peer review of "Chemical and Enzymatic Characterization of Leaves from Spanish Table Olive Cultivars"

_foods, 2022, doi:10.3390/foods11233879_

Round 1

Reviewer 1 Report

This article entitled “Chemical and enzymatic characterization of leaves from Spanish table olive cultivars” described the available data of chemical and enzymatic characterization on leaves from the processing of table olives. This topic is interested; however, it needs minor revisions.

l  The language should be further polished due to it is not precise refinements and too much indirect descriptions are present.

l  The others are fine.

Author Response

Response to Reviewer 1 Comments

Point 1: This article entitled “Chemical and enzymatic characterization of leaves from Spanish table olive cultivars” described the available data of chemical and enzymatic characterization on leaves from the processing of table olives. This topic is interested; however, it needs minor revisions.

Response 1: We want to thank the reviewer for the revision.

Point 2: The language should be further polished due to it is not precise refinements and too much indirect descriptions are present. The others are fine.

Response 2: The manuscript was revised by a native-English translator before submission.

Reviewer 2 Report

In this study, the concentration of polyphenols, triterpenic acids, sugars and enzymatic activities (polyphenol oxidase, peroxidase, β-glucosidase and esterase) have been determined in the leaves of the olive tree (Olea europaea L.) of cvs. Aloreña, Cacereña, Empeltre, Hojiblanca, Manzanilla, Verdial, Gordal and Morona.

This study is very interesting and well written and compiled. The content flow is smooth and easy to read. The experiments were executed in a well-designed manner with experimental controls. The aim and objectives of the study are well addressed.

Author Response

Response to Reviewer 2 Comments

Point 1: In this study, the concentration of polyphenols, triterpenic acids, sugars and enzymatic activities (polyphenol oxidase, peroxidase, β-glucosidase and esterase) have been determined in the leaves of the olive tree (Olea europaea L.) of cvs. Aloreña, Cacereña, Empeltre, Hojiblanca, Manzanilla, Verdial, Gordal and Morona. This study is very interesting and well written and compiled. The content flow is smooth and easy to read. The experiments were executed in a well-designed manner with experimental controls. The aim and objectives of the study are well addressed.

Response 1: We want to thank the reviewer for the revision.

Reviewer 3 Report

The paper turns out to be interesting, results of the contents of phenolic compounds, enzymes, and sugars of leaves of different varieties of olive trees are presented. Some aspects of the discussion can be improved. Below I send you observations resulting from the review of the work.

Materials and methods

Lines 108-113. Would it be possible for them to include the geographical coordinates of the collection areas?

Line 130, 138, 144, 149, 161, 176, 179, 182, 183, 188, 192, 196, 202, 203, 210, 220. Write degrees Celsius correctly.

Lines 293-295. What types of enzymes are involved in the hydrolysis of phenolic compounds?

Line 313. Why is an inversion in the concentration of triterpenes observed in fresh olives?

Line 317. 1 sample is sample 1? Change sample for orchard.

Lines 320-321. Indicate why mannitol is a promising source of bioactive compounds. Is it their precursor? which?

Line 329. Does the low concentration of oleuropein have to do with the rainy season in the collection area or with water stress?

Figure 1 shows the results observed for the different compounds with respect to the different seasons of the year; variations are observed; however, the discussion is not deepened, it is clear that the season of the year affects the concentrations of the compounds being measured, why are these variations due? How do climatic conditions affect the tree, which, in turn, affect the concentrations of the compounds in the leaves? Was the leaf harvest before or after the olive harvest? would this affect the concentration of compounds in the leaves?

Lines 365-366. Panel A and Panel B are Figures 2A and 2B?

In figure 2 it is mentioned that the data presented are the means of the duplicates and that the vertical bars are the standard deviation; how did they do it? To generate the error bars, at least three replicates are needed to obtain a reliable standard deviation.

Figure 3. Homogenize, change pans A and B to 3A and 3B

Author Response

Response to Reviewer 3 Comments

Point 1: The paper turns out to be interesting, results of the contents of phenolic compounds, enzymes, and sugars of leaves of different varieties of olive trees are presented. Some aspects of the discussion can be improved. Below I send you observations resulting from the review of the work.

Response 1: We want to thank the reviewer for the revision. All the observations remarked by the reviewer have been modified in the manuscript and the discussion has been improved.

Point 2: Materials and methods

Lines 108-113. Would it be possible for them to include the geographical coordinates of the collection areas?

Response 2: The reason why the geolocation of the orchards has not been included is that we only know the geographical coordinates of the samples collected from the experimental fields of the Instituto de la Grasa (CSIC) (Seville) and IFAPA Alameda del Obispo (Córdoba). The rest of the samples were supplied by table olive processing industries, and for the privacy policy, they prefer to remain anonymous.

Point 3: Line 130, 138, 144, 149, 161, 176, 179, 182, 183, 188, 192, 196, 202, 203, 210, 220. Write degrees Celsius correctly.

Response 3: Celsius degree writing has been corrected

Point 4: Lines 293-295. What types of enzymes are involved in the hydrolysis of phenolic compounds?

Response 4: The description of enzymes involved in the hydrolysis of phenolic compounds is described in the introduction, lines 77-98. However, the enzymatic activities and several references have been introduced in the text for a better understanding.

Point 5: Line 313. Why is an inversion in the concentration of triterpenes observed in fresh olives?

Response 5: An explanation has been included in the text. This phenomenon has been studied by Romero et al. (2017) finding that the concentrations of maslinic and oleanolic acids do not accumulate in the same proportions in the leaves as in the fruits of the same tree. The biochemical explanation of biosynthesis is unknown to us.

Point 6: Line 317. 1 sample is sample 1? Change sample for orchard.

Response 6: The names of the samples have been renamed.

Point 7: Lines 320-321. Indicate why mannitol is a promising source of bioactive compounds. Is it their precursor? which?

Response 7: An explanation and a new reference (Dai, et al., 2017) have been added in the text.

Point 8: Line 329. Does the low concentration of oleuropein have to do with the rainy season in the collection area or with water stress?

Figure 1 shows the results observed for the different compounds with respect to the different seasons of the year; variations are observed; however, the discussion is not deepened, it is clear that the season of the year affects the concentrations of the compounds being measured, why are these variations due? How do climatic conditions affect the tree, which, in turn, affect the concentrations of the compounds in the leaves?

Response 8: In this work, the olive orchards were cultivated under standard cultural practices and trees were irrigated ad libitum to avoid water stress (line 116). We can assume that the amount of rain, the weather, the time of flowering, or the time of fruit ripening influences the synthesis of polyphenols in the leaf (as discussed in the text), but we cannot support this hypothesis with the data obtained in this work. However, the data obtained correlated the lower values of phenolic compounds with higher polyphenol oxidase activity (line 400).

Point 9: Was the leaf harvest before or after the olive harvest? would this affect the concentration of compounds in the leaves?

Response 9: The leaves corresponding to the month of October were collected before the fruit harvesting, and in the month of December, the fruit was already collected. Between both sampling points, there are no significant differences regarding the phenolic content of the leaves for Manzanilla and Gordal, and a slight increase for Hojiblanca was noticed, which is not conclusive. A clarification has been introduced in the text (lines 339-341).

Point 10: Lines 365-366. Panel A and Panel B are Figures 2A and 2B?

Response 10: Panel A and B have been removed from the text.

Point 11: In figure 2 it is mentioned that the data presented are the means of the duplicates and that the vertical bars are the standard deviation; how did they do it? To generate the error bars, at least three replicates are needed to obtain a reliable standard deviation.

Response 11: Olive leaf samples were collected by duplicates and the analysis was performed also by duplicates, so the standard deviation is reliable. Standard deviations were calculated using Statistica software 10.0 (StatSoft, Inc., Tulsa, OK, USA).

 Point 12: Figure 3. Homogenize, change pans A and B to 3A and 3B

Response 12: The way of naming Figures 2 and 3 has been homogenized.

Reviewer 4 Report

I reviewed the article titled, Chemical and enzymatic characterization of leaves from Spanish table olive cultivars. This study has no novelty. There are many studies available in literature dealing with phenolic analysis of table olive. Authors said that the cultivar is Spanish cultivar. This way it will not contribute to the field. Identification of phenolic compounds and other Triterpenic acids and Sugars based on different months/seasons is of very old concept. For examples, https://doi.org/10.1016/j.foodchem.2020.126626, https://doi.org/10.1039/C9FO00698B, https://doi.org/10.3390/antiox11030558 and     https://doi.org/10.1016%2Fj.btre.2019.e00347  discussed the phenolic profile of table olive leaves. Moreover, authors failed to show the application of these compounds for example, as antimicrobial or antioxidant or food application. This study is also incomplete. There is no purification of phenolic extracts.

Figure caption should come below the Figure

Discussion must be improved

References are not according to the journal format.

Author Response

Response to Reviewer 4 Comments

Point 1: I reviewed the article titled, Chemical and enzymatic characterization of leaves from Spanish table olive cultivars. This study has no novelty. There are many studies available in literature dealing with phenolic analysis of table olive. Authors said that the cultivar is Spanish cultivar. This way it will not contribute to the field. Identification of phenolic compounds and other Triterpenic acids and Sugars based on different months/seasons is of very old concept. For examples, https://doi.org/10.1016/j.foodchem.2020.126626 , https://doi.org/10.1039/C9FO00698B , https://doi.org/10.3390/antiox11030558  and     https://doi.org/10.1016%2Fj.btre.2019.e00347   discussed the phenolic profile of table olive leaves.

Response 3: We want to thank the reviewer for the comments. As commented, there are many studies in the literature about phenolic content in table olives, and also in olive oil. Although several researchers have studied the phenolic concentration in olive leaves, all the studies have been carried out with olive leaves of varieties destined for the production of olive oil.

Regarding the examples shown by the reviewer, https://doi.org/10.1016/j.foodchem.2020.126626 studied the leaves from olive oil varieties Arbequina, Picual, Royal and 3 wild varieties; In the work https://doi.org/10.1039/C9FO00698B, commercial leaf samples were studied, without specifying the variety or origin. In the same way, in the publication, https://doi.org/10.3390/antiox11030558,  the leaves from olive oil varieties Arbequina, Chaglot Real, Frantoio, Picual, Koroneiki and Sikitita were characterized. And finally, the study https://doi.org/10.1016%2Fj.btre.2019.e00347 didn´t specify the variety of the leaves, and only compared the phenolic profile after different extraction methods. In our work, this is the first time that olive leaves of varieties such as Aloreña, Cacereña, Empeltre, Hojiblanca, Manzanilla, Verdial, Gordal and Morona cultivars, used for the production of table olives, not for olive oil, have been characterized.

The study of the phenolic composition in different seasons has been carried out by several authors, as discussed in the manuscript, but its correlation with the enzymatic activities related to the degradation of phenols, especially with polyphenol oxidase, has not been studied before. Very few authors have studied the enzymatic activity of the olive leaf. The novelty of this study is the exhaustive analysis of enzymatic activities carried out in a wide range of olive varieties never study before.

Point 2:  Moreover, authors failed to show the application of these compounds for example, as antimicrobial or antioxidant or food application. This study is also incomplete. There is no purification of phenolic extracts.

Response 3: The goal of this work was to generate knowledge about the chemical and enzymatic characterization that bioactive substances can undergo in leaves from the main Spanish olive cultivars used for table olive elaboration, and to understand the mechanisms that can affect the profiles of these active compounds, not previously studied.

Most of the studies carried out with olive leaves have aimed to produce extracts rich in bioactive compounds, particularly phenolic compounds, with the objective of being commercialized by companies for food additives or nutraceutical purposes. The purification of phenolic extracts is not the objective of this work. The goal is to characterize the content of bioactive compounds in olive leaves in order to select those varieties with a higher concentration for the elaboration of dried leaves for use in infusions and beverages healthy for the consumer, highly demanded by the food sector. In fact, the knowledge of the enzymatic activities in the leaves, never before studied, could modify the original phenolic profile of the olive leaf with the purpose of maximizing the content of bioactive compounds in these beverages. The optimization of the olive beverage production process will be addressed in future studies, and then the antimicrobial and antioxidant activities will be studied.

Point 3: Figure caption should come below the Figure

Response 3: The figure captions have been moved below the Figures.

Point 4: Discussion must be improved

Response 3: All the observations remarked by the reviewers have been modified in the manuscript and the discussion has been improved.

Point 5: References are not according to the journal format.

Response 3: References have been modified to the journal format.

Round 2

Reviewer 4 Report

Although the manuscript has less contribution to the field, authors reply to my questions are not convincing. Thus, I must not to change my recommendation as "REJECT"